# Synonymous alterations of cancer-associated *Trp53* CpG mutational hotspots cause fatal developmental jaw malocclusions but no tumors in knock-in mice

Richard J. Epstein[1,2]*, Frank P. Y. Lin[1,3], Robert A. Brink[1,2], James Blackburn[1,2]

**1** University of New South Wales, St Vincent's Hospital Campus, Sydney, Australia, **2** Garvan Institute of Medical Research, Sydney, Australia, **3** Centre for Clinical Genomics, The Kinghorn Cancer Centre, Sydney, Australia

\* r.epstein@unsw.edu.au

**Data Availability Statement:** All relevant data are within the paper and its Supporting Information files.

## Abstract

Intragenic CpG dinucleotides are tightly conserved in evolution yet are also vulnerable to methylation-dependent mutation, raising the question as to why these functionally critical sites have not been deselected by more stable coding sequences. We previously showed in cell lines that altered exonic CpG methylation can modify promoter start sites, and hence protein isoform expression, for the human *TP53* tumor suppressor gene. Here we extend this work to the in vivo setting by testing whether synonymous germline modifications of exonic CpG sites affect murine development, fertility, longevity, or cancer incidence. We substituted the DNA-binding exons 5–8 of *Trp53*, the mouse ortholog of human *TP53*, with variant-CpG (either CpG-depleted or -enriched) sequences predicted to encode the normal p53 amino acid sequence; a control construct was also created in which all non-CpG sites were synonymously substituted. Homozygous *Trp53*-null mice were the only genotype to develop tumors. Mice with variant-CpG *Trp53* sequences remained tumor-free, but were uniquely prone to dental anomalies causing jaw malocclusion (p < .0001). Since the latter phenotype also characterises murine Rett syndrome due to dysfunction of the trans-repressive MeCP2 methyl-CpG-binding protein, we hypothesise that CpG sites may exert non-coding phenotypic effects via pre-translational cis-interactions of 5-methylcytosine with methyl-binding proteins which regulate mRNA transcript initiation, expression or splicing, although direct effects on mRNA structure or translation are also possible.

## Introduction

The CpG dinucleotide represents a masterstroke of adaptive evolution [1–3], combining as it does the stabilization of Watson-Crick base pairs via its triple hydrogen-bonded structure [4–6] with destabilization of sequence fidelity via a susceptibility to cytosine methylation that in turn permits mutagenic deamination to thymine [7–9]. Even without this adaptive role, the informatic content of CpG sites exceeds that of amino acid coding alone, given that

**Funding:** The funders had no role in study design, data collection and analysis, decision to publish, or preparation of the manuscript.

**Competing interests:** The authors have declared that no competing interests exist.

methylation of CpG-cytosines in canonical B-DNA regulates conformational changes in nucleosomes which affect gene transcription, duplication or deletion [10, 11]. From a mechanistic viewpoint, pre-transcriptional intragenic (gene body) CpG modification [12–14] by methylation writers [15] (methyltransferases) enables the site-specific DNA binding [16, 17] of trans-acting readers (methylcytosine-binding proteins, MBPs, such as MeCP2 or MBD2 [18–20])–germline dysfunctions of which may cause neurological disorders [21–23]–or else erasure by DNA methylation editors (TET proteins [24]). This epigenetic plasticity of 5-methylcytosine [25] (5mC; sometimes termed the fifth base of DNA [26]) has been proposed to be the basis of an alternate genomic code that diversifies gene function [27].

We have reported correlations between amino acid functionality and conservation of CpG-containing codons [28, 29] when compared with synonymous non-CpG codons such as may have arisen via methylation-dependent CG→TA transitions [30]. The importance of conserved exonic CpG sites is further implied by the biology of diseases like cancer in which mutations of such sites play a central role in pathogenesis [31]. It therefore seems likely that the letters of the genetic code contain meanings not limited to polypeptide synthesis. Yeast studies confirm that synonymous mutations yield deleterious phenotypes with similar efficiency to nonsynonymous mutations [32], reflecting DNA base-specific modifications of mRNA expression [33] or splicing [34].

*TP53*, a pivotal cell-cycle regulator [35–37], is the most mutated gene in human cancer [38]. CpG dinucleotides in exons 5–8 of this gene are constitutively methylated [39], predisposing to age- or carcinogen-induced mutations [40] that alter DNA binding by the encoded p53 proteins, and so dysregulate growth control [41, 42]. Missense mutations of this kind may not only abrogate tumor suppression, but also act in a dominant-negative manner to drive cancer progression [43]. Hence, as well as permitting carcinogenesis [44], CpG-related *TP53* mutations often confer behavioral aggressivity on tumors such as breast [45], colorectal [46], ovarian [47], prostate [48] and lung cancers [49].

Exons 5–8 of the human *TP53* and mouse *Trp53* genes encode the critical p53 DNA-binding domain [50]. These orthologous exons exhibit high (> 90%) sequence homology, including > 95% CpG site retention, and 100% conservation of the main cancer-predisposing hydrophilic *CGX*-encoded arginine residues at positions 158, 175, 248, 273 and 282 [51]. Perfect conservation of these codon-specific amino acids across mammalian, fish and amphibian species [52] supports the validity of using mouse synonymous *Trp53* knock-ins to infer human *TP53* functions, as implied by domain-swapped human p53 knock-in (Hupki) mice [53, 54], as well as by the tumorigenicity of murine *Trp53* mutations homologous to Li-Fraumeni-type *TP53* codon 175/273 mutations [55]. The strongly conserved germline CpG sites of human *TP53* are therefore fairly regarded as a hypermutable genomic Achilles heel for somatic cancer causation and progression (S1 Fig) [50].

We have shown in earlier work involving synonymous CpG-substituted human *TP53* cDNAs that abnormal demethylation of a *TP53* CpG site in exon 5 activates an intron 4 promoter, leading to production of a truncated protein isoform [56]. Such p53 isoforms are reported to exert anti-apoptotic effects that inhibit the cell-regulatory functions of full-length p53 [57–60], as could manifest as developmental malformations [61]. We have now used CRISPR/Cas9 to create knock-in mice expressing synonymous variable-CpG (vc) *Trp53* exons 5–8, and thus to test whether germline expression of one or both of these CpG extremes causes embryogenetic defects, impaired fertility, shortened survival, or altered susceptibility to spontaneous cancers, despite the predicted absence of change in the amino acid sequence of p53 proteins encoded.

## Materials and methods

### Synonymously substituted sequence design

In addition to knockout constructs (see below), three synonymous knock-in sequences were synthesised: (i) a putative ultra-stable (CpG-) exon 5–8 cDNA in which all 22 wild-type *Trp53* CpG sites were replaced by non-CpG dinucleotides predicted to encode the original amino acid sequence; (ii) a putative hyper-mutable (CpG+) cDNA in which existing CpG sites were retained, but a further 72 CpG sites were synonymously inserted; and (iii) a control non-CpG synonymous (NCpGS) mutated cDNA in which wild-type CpG sites remained unchanged, but 181 non-CpG sequences were altered in a manner predicted to leave the p53 amino acid structure intact. For the ultra-stable CpG- construct, all 22 CpG sites proved to be synonymously replaceable (S1 Table), re-emphasising the evolutionary question over their conservation.

For the hyper-mutable CpG+ construct, divergent CpG sites were substituted even when this did not reduce the number of sites–for example, ACG/GAG became ACC/GAG–to test further whether the precise location of a CpG base pairing contributes to its function. In addition, CpG adjacent sequences were changed if this could alter the evolutionarily selectable effects of spontaneous mutation; hence, the CGA trinucleotide was substituted by CGG, since CGA mutation to TGA causes a stop codon, whereas CGG to TGG specifies tryptophan (S2 Table).

### Knock-in mice

All *Trp53* mutant mouse lines were produced by the Mouse Engineering Garvan/ABR (MEGA) Facility (Moss Vale/Sydney, Australia) using CRISPR/Cas9 gene targeting in C57BL/6J mouse embryos using standard molecular and animal husbandry techniques [62, 63]. Two single guide RNAs (sgRNAs) targeted Cas9 cleavage 108 bp 5' of exon 5 (ACCATTGGACGCCC TCG*CAG<u>TGG</u>) and 14 bp 3' of exon 8 (GAGGTACGCAGGCGGGAG*CCA<u>AGG</u>) of *Trp53* (* = Cas9 cleavage site, underlined = proto-spacer-associated motif (PAM)). Three 3980 bp homologous recombination (HR) substrates were synthesized in pUC57 plasmid (Genscript, Piscataway, NJ), each of which included 1000 bp 5' and 1500 bp 3' homology arms either side of the two Cas9 target sequences and a G>C mutation in the 3' base of each of the PAMs. The remainder of the substrates carried sequences corresponding to the intervening regions of *Trp53* gene but contained the specific base changes in exons 5–8 (S3–S5 Figs) designed to produce the CpG+, CpG- and NCpGS lines. In each case, a solution consisting of the two sgRNAs (15 ng/μl each), purified double stranded HR substrate plasmid DNA (2 ng/μl) and full length, polyadenylated *S.pyogenes* Cas9 mRNA (30 ng/μl) was prepared and microinjected into the nucleus and cytoplasm of C57BL/6J zygotes. Microinjected embryos were cultured overnight and those that underwent cleavage introduced into pseudo-pregnant foster mothers. Pups were screened by PCR to detect homologous recombination of each HR substrate into the *Trp53* alleles and founder animals crossed with C57BL/6J mice to establish the three *Trp53* knock-in lines. In addition, a mouse in which the sequences between the two Cas9 target sequences had been deleted (and hence, exons 5–8 removed) was used to establish a *Trp53* knockout line.

### Ethics and safety

Mice were bred at Australian BioResources (ABR; MossVale, NSW, Australia) and housed in specific pathogen-free conditions. All animal studies were approved and conducted in compliance with the guidelines set by the Garvan/St.Vincent's Animal Ethics Committee. Mice in

this study were monitored weekly for weight changes, signs of discomfort, or poor health; where humanely indicated by such endpoints (e.g., signs of pain, inability to reach food, or $\geq 20\%$ weight loss), mice were sacrificed by carbon dioxide asphyxiation. No anaesthesia or analgesia was carried out. Pre-euthanasia causes of death are listed in Table 1. At any one time, 10–12 mice were analysed per genotype to ensure interpretability of observations.

## Results

Using the above approach we produced 1458 transgenic mice, comprising:

• 470 *Trp53*-null mice (from heterozygous knockout line ID 5555);

• 732 variable-CpG (vc)-*Trp53* mice, in turn comprising:

○ 322 *Trp53* CpG- mice, line ID 5801

○ 410 *Trp53* CpG+ mice, line ID 5774

• 256 synonymous non-CpG-mutated NCpGS *Trp53* mice, line ID 5775

The phenotypic observations from these knock-in mice are summarised in Table 1. As expected, tumor formation was evident in homozygous *Trp53* knockout mice (together with

**Table 1. Observed phenotypes in transgenic mice.**

| Animal no. | Short line name | Genotype | Sex (M/F) | Survival (weeks) | Post mortem phenotype |
|---|---|---|---|---|---|
| 115 | | | M | 22 | Tumor |
| 161 | | | M | 17 | Tumor |
| 317 | | | M | 28 | Tumor |
| 131 | | Homozygous *Trp53* Δ | M | 19 | Abdominal bloating |
| 160 | | | M | 15 | Facial abscess |
| 318 | | | M | 16 | Squashed |
| 96 | *Trp53* Δ | | F | 14 | Dystocia |
| 167 | | | F | 12 | Dystocia |
| 291 | | Heterozygous *Trp53* Δ | F | 10 | Dystocia |
| 395 | | | F | 24 | Dystocia |
| 337 | | | M | 3 | Hydrocephalus |
| 332 | | | F | 30 | Bite wounds |
| 324 | | | M | 4 | Unknown |
| 4 | *Trp53* (NCpGS) | Heterozygous *Trp53* (NCpGS) | F | 22 | Dystocia |
| 65 | | | F | 18 | Dystocia |
| 116 | | | F | 19 | Vaginal septum |
| 182 | | | F | 18 | Not documented |
| 23 | | | F | 12 | Malocclusion |
| 38 | *Trp53* (CpG-) | Heterozygous *Trp53* (CpG-) | F | 3 | Malocclusion |
| 243 | | | F | 6 | Malocclusion |
| 133 | | Homozygous *Trp53* (CpG-) | M | 6 | Malocclusion |
| 271 | *Trp53* (CpG+) | Heterozygous *Trp53* (CpG+) | M | 5 | Malocclusion |
| 199 | | | M | 5 | Malocclusion |
| 152 | | Homozygous *Trp53* (CpG+) | M | 5 | Malocclusion |
| 117 | | | M | 3 | Post-wean demise |

Mice with *Trp53* knockout and/or CRISPR/*Cas9* knock-in using either vc-*Trp53* (CpG- or CpG+) or synonymous non-CpG *Trp53* (NCpGS) constructs were observed for developmental (e.g., congenital malformation) or adult (e.g., tumor) phenotypes.

some other late-onset phenotypes suspicious for tumor growth, such as abdominal bloating and facial abscess, though the latter phenotype can also sometimes indicate subclinical incisor growth defects [64]; see below). In contrast, no tumors were detected in any subset (i.e., either CpG+ or CpG-) of vc-*Trp53* knock-in mice, nor in the NCpGS synonymous non-CpG mutation control. A high frequency of dystocia in *Trp53* homozygous knockout mice was also recorded; to the best of our knowledge, dystocia has not previously been reported as a feature of germline p53 loss of function, although it has been reported in double-knockout FasL-/p53-mice [65].

Since the expected spontaneous frequency of malocclusion in C57BL/6J mice is 0.05%, i.e., one in 2000 [64], an unanticipated finding of this study was the clustering of six jaw malocclusion defects with vc-*Trp53* genotypes (Fig 1), compared to only one such malocclusion in non-vc-*Trp53* mice; conversely, vc-*Trp53* mice incurred only one other phenotype (labelled post-wean demise, with no other details), whereas non-vc-*Trp53* mice developed 17 other, non-malocclusion, phenotypes ($\chi^2$ with Yates correction = 12.3, p < .0001). Malocclusion was associated with premature mortality in vc-*Trp53* mice, with survival averaging only 5 +/- 0.4 weeks, presumably due to severe feeding problems (e.g., see Fig 1A); the average survival of other lethal phenotypes in this series was 17 +/- 1.7 weeks. The only non-vc-*Trp53* mouse with malocclusion (in the NCpGS genotype) survived for 12 weeks, i.e., longer than in the affected vc-*Trp53* mice.

A notable feature of the vc-*Trp53*-associated malocclusion cluster is that it was phenotypically evident with both CpG- and CpG+ vc-*Trp53* genotypes, as well as with heterozygous and homozygous gene dosages. This could suggest a dominant effect caused by aberrations that occur in the absence of evolutionarily conserved (i.e., correct) spatial CpG positioning, as discussed further below.

Beyond the above-noted effects on dental/jaw morphology, and the lack of any detectable effect on spontaneous tumor frequency, significant negatives of the study included no evident effect of vc-*Trp53* (or NCpGS) genotypes on mouse fertility or longevity.

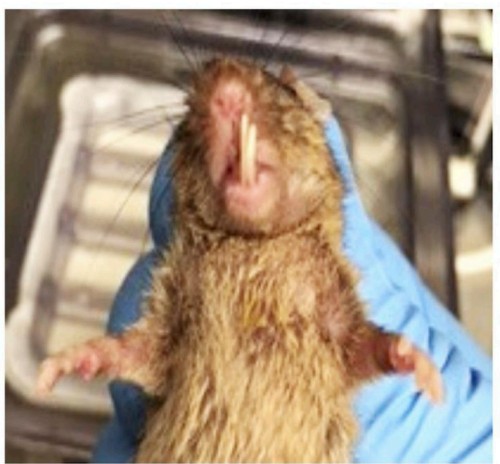 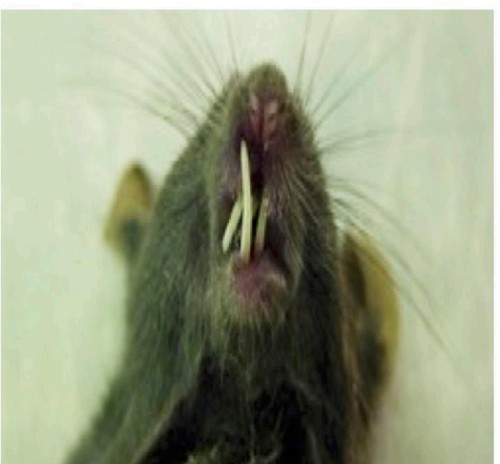

**A** **B**

**Fig 1. Typical examples of jaw malocclusion.** A, Whole-mouse photograph showing the dysmorphic appearance of the incisors, as well as associated rapid malnutrition due to feeding difficulties. B, Detailed photo of incisor overgrowth defect underlying malocclusion and feeding problems.

## Discussion

Consistent with abnormal p53-dependent apoptosis, transgenic mice with p53 anomalies can develop upper incisor tooth malformations [66, 67]. Incisor defects likewise occur in rodent models of Rett syndrome, and lead to jaw malocclusion [68]; in this context, loss of normal MeCP2-dependent trans-repression [69] dysregulates mRNA splicing [70–72]. Relevant to this, the ability of wild-type MeCP2 to direct normal splicing events depends on exon recognition which is based in turn on correct spatial recruitment by intragenic methylation sites [73]. These observations are consistent with the hypothesis raised by ourselves and others that evolutionary CpG conservation could in part reflect a teratogenesis-mediated selection pressure which is operative in utero [74–76].

Hence, a plausible evo-devo view [77] of our results is that disrupting the conserved intragenic CpG structure of *Trp53* –in this study, either by synonymously removing all 22 exonic CpG sites, or else by inserting 72 synonymous CpG sites–deranges the topology of site-specific MBP (e.g., MeCP2 or MBD2) interactions with *Trp53*, leading to loss of pro-apoptotic gene function in the developing tooth bed [78, 79], e.g., by splice-dependent [80] or chromatin-based *trans*-repression mechanisms [81, 82]. Consistent with this, *TP53* is downregulated in proliferative dental pathologies such as odontogenic keratocysts and ameloblastomas [83]. Moreover, malregulation of the p53-interactive homolog p63 [67, 84, 85] is firmly implicated in tooth and craniofacial defects [86–88].

A question prompted by these findings relates to the fate of organisms with more extensive– e.g., genome-wide, rather than *Trp53*-specific–synonymous exonic variant-CpG genotypes. We speculate that such widespread CpG change could prove embryonic lethal, since it may cause intragenic dysregulatory effects on a scale equating to complete knockdown of all MBPs. Conversely, the negative finding of this study–viz., that no vc-*Trp53* knock-in mouse lineages developed spontaneous tumors, unlike their loss-of-function *Trp53* knockout controls–suggests that in vivo applications of this cellular approach, whether germline or somatic, may be safe.

It should be noted that there are additional explanations for these results which are independent of MBPs. For example, one possibility is that alterations of intragenic CpG sites could create or remove as-yet-unrecognised alternative transcription initiation sites that in turn alter function of the encoded protein [89, 90]. A second possibility is that gain or loss of CpG sites alters mRNA secondary structure and (hence) protein folding [91], perhaps via changes in base stacking and Z-RNA formation [92, 93] due to modification of CpG step number or position [94, 95]. Yet another possibility is that altered intragenic CpG content could affect ribosomal translational pausing with consequent downstream effects on protein translation efficiency or subdomain folding [96].

There remain several other important limitations of this study. The short-term mouse knock-in system is not a definitive in vivo test, as it does not exclude different results when assessed over other timescales or conditions [97]–such as over a human lifespan or, more crucially, as could relate to evolutionary CpG-dependent speciation events selected by environmental change [98–101]. Moreover, the finding that CpG-enriched and CpG-depleted constructs are both associated with malocclusion, as are homozygotes and heterozygotes, complicates interpretation; a qualitative explanation mandating a perfectly correct spatial arrangement of methyl-binding proteins is possible, and consistent with the tight evolutionary conservation of CpG sites, and the similar Rett phenotype. Finally, it remains unclear why this reported experimental phenotype is so restricted to orodental tissues, and how this might relate to a tissue-specific loss or change of p53 function, although intermediary effects on malocclusion-specific genes [102] or dental proteins [103, 104] are by no means excluded in the etiopathogenesis.

## Conclusion

The association between synonymous CpG-variant *Trp53* sequences and maldevelopment reported here supports, albeit indirectly, a growing body of epigenetic evidence favoring a CpG phenotype, though it is unclear whether this is due to pre-translational methylation-dependent interactions with DNA-binding trans-acting factors, or to MBP-unrelated changes in mRNA processing, structure, or translation. In other respects, the study suggests that synonymous replacement of conserved exonic CpG sites is well tolerated in unstressed mammalian tissues.

## Supporting information

**S1 Fig. Relationship between amino acid site-specificity of sporadic carcinogenic mutations (above) and evolutionary rate (Ka/Ks, below: red) in *TP53*.** Sequences were downloaded from NCBI Entrez Gene (http://www.ncbi.nlm.nih.gov/Entrez/Gene), and homolog data in XML format from NCBI Homolo-Gene database (ftp://ftp.ncbi.nih.gov/pub/HomoloGene/). Mutation data were downloaded from the Human Gene Mutation Database. K-estimator 6.1 (with window size of 33 codons and step size of 10 codons using Kimura 2-parameter method) and PAML 3.15 with yn00 model were used for evolutionary rate calculations. Orthologous gene pairs between human and mouse, together with their synonymous substitution (Ks), nonsynonymous substitution rate (Ka), and their ratio (Ka/Ks), were thus isolated. The Ka/Ks evolutionary rate for *TP53* CpG sites in exons 5–8 was shown to approach zero, consistent with high negative selection pressure, with these same (functionally important) germline sites closely corresponding to those undergoing somatic mutation in tumors.
(TIF)

**S2 Fig. Illustration of mutagenesis strategy based on earlier in vitro studies using human *TP53* cDNA constructs.** *A*, Representation of synonymous mutations introduced into cDNA constructs, exons 5–8. Open circles–ancestral CpG sites. Red symbols–additional synonymous CpG sites (CGN, NCG, NNC/GNN). The first (wild-type) cDNA lacks the three introns normally bridging exons 5–8; the second construct, with synonymous losses of wild-type CpG sites, is labeled stable; the third construct, to which synonymous CpG-containing sites have been added, is labeled missense. *B*, Hypothetical phenotypic effects as potential downstream somatic consequences of altered germline *TP53* mutation frequencies secondary to the synonymous changes.
(TIF)

**S3 Fig. Synonymous mutagenised *Trp53* CpG- sequence changes.** 22 CpG sites highlighted in yellow are replaced as shown by synonymous green-highlighted bases.
(TIF)

**S4 Fig. Synonymous mutagenised *Trp53* CpG+ sequence changes.** In addition to the retained wild-type CpG sites (highlighted in purple), 72 new CpG sites are created by synonymous replacement by the bases highlighted in either blue or yellow.
(TIF)

**S5 Fig. Mouse exons 5–8, all non-CpG-synonymous (NCpGS) versus WT.** CpG sites highlighted in yellow remain conserved in both sequences. Green-highlighted bases in the NCpGS sequence represent all other (181) possible synonymous nucleotide changes.
(TIF)

**S1 Table. Exon-specific synonymous base changes in CpG- vc-*Trp53* mice.** CpG sites are highlighted.
(TIF)

**S2 Table. Exon-specific synonymous base changes in CpG+ vc-*Trp53* mice.** CpG sites are highlighted.
(TIF)

## Acknowledgments

We thank the Garvan Institute ABR and GMG facilities, including Kevin Taylor and colleagues, for animal husbandry and genotyping, and Dr Yongzhong Zhao for data in support of S1 Fig.

## Author Contributions

**Conceptualization:** Richard J. Epstein.

**Data curation:** Frank P. Y. Lin, Robert A. Brink.

**Formal analysis:** James Blackburn.

**Funding acquisition:** Richard J. Epstein.

**Investigation:** Richard J. Epstein, Robert A. Brink.

**Methodology:** Robert A. Brink, James Blackburn.

**Project administration:** Richard J. Epstein, Frank P. Y. Lin.

**Resources:** Robert A. Brink.

**Supervision:** Richard J. Epstein, Robert A. Brink.

**Validation:** Frank P. Y. Lin, James Blackburn.

**Writing – original draft:** Richard J. Epstein.

**Writing – review & editing:** Richard J. Epstein, Frank P. Y. Lin, Robert A. Brink, James Blackburn.

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
