## [Decision Letter · Decision Letter 0]

6 Mar 2023

PONE-D-22-34300Synonymous alterations of cancer-associated Trp53 CpG mutational hotspots 

cause fatal developmental jaw malocclusions but no tumors in knock-in micePLOS ONE

Dear Dr. Epstein,

Thank you for submitting your manuscript to PLOS ONE. After careful consideration, we feel that it has merit but does not fully meet PLOS ONE’s publication criteria as it currently stands. Therefore, we invite you to submit a revised version of the manuscript that addresses the points raised during the review process.

 Please submit your revised manuscript by Apr 20 2023 11:59PM. If you will need more time than this to complete your revisions, please reply to this message or contact the journal office at plosone@plos.org. Please include the following items when submitting your revised manuscript:A rebuttal letter that responds to each point raised by the academic editor and reviewer(s). You should upload this letter as a separate file labeled 'Response to Reviewers'.A marked-up copy of your manuscript that highlights changes made to the original version. You should upload this as a separate file labeled 'Revised Manuscript with Track Changes'.An unmarked version of your revised paper without tracked changes. You should upload this as a separate file labeled 'Manuscript'.If applicable, we recommend that you deposit your laboratory protocols in protocols.io to enhance the reproducibility of your results. Protocols.io assigns your protocol its own identifier (DOI) so that it can be cited independently in the future. For instructions see: https://journals.plos.org/plosone/s/submission-guidelines#loc-laboratory-protocols. Additionally, PLOS ONE offers an option for publishing peer-reviewed Lab Protocol articles, which describe protocols hosted on protocols.io. Read more information on sharing protocols at https://plos.org/protocols?utm_medium=editorial-email&utm_source=authorletters&utm_campaign=protocols.

We look forward to receiving your revised manuscript.

Kind regards,

Sumitra Deb, PhD

Academic Editor

PLOS ONE

Journal Requirements:

2. In your Methods section, please provide additional information on the animal research and ensure you have included details on : (1) methods of sacrifice (2) methods of anesthesia and/or analgesia, and (3) the cause of death for animals before they meet the criteria for euthanasia.

3. Thank you for stating the following in the Acknowledgments/ Funding Section of your manuscript: 

This work was supported by a donation from the family of Mr Frank Wolf to the St Vincent’s Curran Foundation. The funders had no role in study design, or in manuscript preparation.

Reviewers' comments:

Reviewer's Responses to Questions

**Comments to the Author**

1. Is the manuscript technically sound, and do the data support the conclusions?

Reviewer #1: Yes

2. Has the statistical analysis been performed appropriately and rigorously? 

Reviewer #1: N/A

3. Have the authors made all data underlying the findings in their manuscript fully available?

Reviewer #1: Yes

4. Is the manuscript presented in an intelligible fashion and written in standard English?

Reviewer #1: Yes

5. Review Comments to the Author

Reviewer #1: The paper describes a interesting experiment: the authors substituted the DNA-binding exons 5-8 of Trp53, the mouse ortholog of human TP53, with variant-CpG (either CpG-depleted or -enriched) sequences predicted to encode the normal p53 amino acid sequence. It is important that a control construct was also created in which all non-CpG sites were synonymously substituted. I do not see any need in additional experiments (many of them can be suggested but I am sure that the paper represents a solid publishable unit).

Comments:

The authors suggested that " that conserved CpG sites exert non-coding phenotypic effects via pre-translational cis-interactions of 5-methylcytosine with methyl-binding proteins which regulate mRNA transcript initiation, expression or splicing". May be, may be not. I think that other possible explanations should be discussed (the discussion is too short anyway). For example, a change of two tandem CpG containing codons of SARS-CoV-2 (that employs the host translation system substantially change properties of the spike protein most likely due to changes in translation :

The Functional Consequences of the Novel Ribosomal Pausing Site in SARS-CoV-2 Spike Glycoprotein RNA.

Postnikova OA, Uppal S, Huang W, Kane MA, Villasmil R, Rogozin IB, Poliakov E, Redmond TM. Int J Mol Sci. 2021 Jun 17;22(12):6490.

or secondary structure effects:

Role of mRNA structure in the control of protein folding. Faure G, Ogurtsov AY, Shabalina SA, Koonin EV. Nucleic Acids Res. 2016 Dec 15;44(22):10898-10911.

Something brief, the paper will benefit, I am sure.

6. PLOS authors have the option to publish the peer review history of their article (what does this mean?). If published, this will include your full peer review and any attached files.

Reviewer #1: No

---

## [Author Response · Author response to Decision Letter 0]

17 Mar 2023

Please see the uploaded Response to Reviewers

---

## [Editor Report · Decision Letter 1]

30 Mar 2023

Synonymous alterations of cancer-associated Trp53 CpG mutational hotspots 

cause fatal developmental jaw malocclusions but no tumors in knock-in mice

PONE-D-22-34300R1

Dear Dr. Epstein,

We’re pleased to inform you that your manuscript has been judged scientifically suitable for publication and will be formally accepted for publication once it meets all outstanding technical requirements.

Kind regards,

Sumitra Deb, PhD

Academic Editor

PLOS ONE
---

## [Editor Report · Acceptance letter]

4 Apr 2023

PONE-D-22-34300R1 

Synonymous alterations of cancer-associated *Trp53* CpG mutational hotspots
cause fatal developmental jaw malocclusions but no tumors in knock-in mice 

Dear Dr. Epstein:

I'm pleased to inform you that your manuscript has been deemed suitable for publication in PLOS ONE. Congratulations! Your manuscript is now with our production department. 

Kind regards, 

on behalf of

Dr. Sumitra Deb 

Academic Editor

PLOS ONE